# Koji Molds for Japanese Soy Sauce Brewing: Characteristics and Key Enzymes

**DOI:** 10.3390/jof7080658

**Published:** 2021-08-13

**Authors:** Kotaro Ito, Asahi Matsuyama

**Affiliations:** Research and Development Division, Kikkoman Corporation, Noda 278-0037, Japan; amatsuyama@mail.kikkoman.co.jp

**Keywords:** koji, soy sauce, *Aspergillus sojae*, *Aspergillus oryzae*, proteolytic enzymes, glutaminase, flavor, umami taste

## Abstract

Soy sauce is a traditional Japanese condiment produced from the fermentation of soybeans, wheat, and salt by three types of microorganisms, namely koji molds, halophilic lactic acid bacteria, and salt-tolerant yeast. The delicate balance between taste, aroma, and color contributes to the characteristic delicious flavor imparted by soy sauce. In soy sauce brewing, protein and starch of the raw materials are hydrolyzed into amino acids and sugars by enzymes derived from koji molds. These enzymatically hydrolyzed products not only directly contribute to the taste but are further metabolized by lactic acid bacteria and yeasts to most of organic acids and aromatic compounds, resulting in its distinctive flavor and aroma. The color of the soy sauce is also due to the chemical reactions between amino acids and sugars during fermentation. Therefore, koji mold, which produces various enzymes for the breakdown of raw materials, is an essential microorganism in soy sauce production and plays an essential role in fermenting the ingredients. In this review, we describe the manufacturing process of Japanese soy sauce, the characteristics of koji molds that are suitable for soy sauce brewing, and the key enzymes produced by koji molds and their roles in the degradation of materials during soy sauce fermentation, focusing on the production of umami taste in soy sauce brewing.

## 1. Introduction

Soy sauce (shoyu in Japanese) is a condiment that forms the foundation of “Washoku” (Japanese cuisine) and is now widely used as an all-purpose seasoning in more than 100 countries worldwide. It is presumed that “hishio,” which is derived from salted fish or grains and originated from China in the 5th century, is the primitive prototype of shoyu and miso (soybean paste) [1]. In 1254, the Buddhist monk Kakushin returned to Japan having learned the method of producing Kinzanji miso from China. He later discovered that the liquid pooling at the bottom of the vats in miso fermentation can be used as a tasty condiment, which was later developed into shoyu. Chinese soy sauce is synthesized from pure soybeans or its mixture with a small amount of wheat, whereas Japanese-style soy sauce uses equal amounts of soybeans and wheat [2]. Many techniques for brewing soy sauce were adopted from sake (Japanese rice wine) production. By the 17th century (end of the Edo era), the basic manufacturing process of soy sauce had been completed and is still mainly utilized at present.

In this review, we introduce the manufacturing process of Japanese soy sauce, focusing on the difference in koji molds used in soy sauce brewing. In addition, since the genome sequence of the koji molds has been clarified, the enzymes involved in the production of umami taste in soy sauce, which have been identified so far, will be introduced in connection with their genes.

## 2. Manufacturing Process of Soy Sauce

Soy sauce is produced by fermenting soybeans, wheat, and salt with koji mold, halophilic lactic acid bacteria, and salt-tolerant yeast. The most typical method of manufacturing soy sauce in Japan is called the “Honjozo” method and has five main processes: (1) treatment of raw materials, (2) making of koji, (3) fermentation of moromi mash and aging, (4) pressing of moromi mash, and (5) refining (Figure 1). In the first step, the soybeans are moistened and cooked under steam pressure to denature the proteins, and the wheat is roasted at 160–180 °C and crushed. Approximately equal amounts of soybean and wheat components are then mixed and inoculated with a pure culture of koji mold. This mixture is incubated for two to three days at 30 °C under high humidity to allow the koji mold to grow throughout the mixture and produce various hydrolyzing enzymes, such as proteolytic and amylolytic enzymes, which are responsible for the decomposition of the raw materials. The dry mash formed is called koji (or soy sauce koji). The quality of soy sauce koji heavily relies on the quality of soy sauce. Therefore, the process of making koji, including the processing of raw materials, is the most crucial step in soy sauce brewing.

Soy sauce koji is then added to a concentrated (22–23%) saline solution to produce Moromi or Moromi mash that is traditionally fermented for 6–8 months. In this process, the entire enzyme-rich koji mixture is used, rather than pure enzymes from the raw materials, to decompose the raw materials and yield amino acids and sugars, which are responsible for the distinctive flavor of Japanese soy sauce. In addition, these amino acids and sugars serve as a nutrient source for lactic acid bacteria and yeast that ferment the raw material. The high salt concentration (17–18%) of soy sauce does not allow the growth of non-salt-tolerant microorganisms. Soon after brewing, the koji mold dies off and is replaced by salt-tolerant or halophilic microorganisms. The first microbes that grow in moromi mash are halophilic lactic acid bacteria.

Halophilic lactic acid bacteria were first isolated from soy sauce moromi mash in 1907 [3]. Initially, it was classified as *Pediococcus soyae*; however, in 1990, it was reclassified as *Tetragenococcus halophilus*. *T. halophilus* is a Gram-positive strain, micrococcus, which often forms a tetrad coccoid, and grows best in an environment containing 5–10% salt. Each strain is highly diverse and exhibits different properties, including amino acid decomposition, sugar-fermentation ability, reducing activity, and organic acid metabolism [3]. *T. halophilus* metabolizes glucose, which is produced by the breakdown of wheat starch by amylolytic enzymes derived from koji mold, into lactic acid and citric acid in soybeans into acetic acid. As a result, the pH of soy sauce moromi is slightly acidic, imparting a slightly sour taste to soy sauce.

After lactic acid fermentation by growing halophilic lactic acid bacteria, two types of salt-tolerant yeast grow, namely major fermenting yeast and late maturation yeast. The former is classified as *Zygosaccharomyces rouxii* and mainly affects the flavor of soy sauce. Meanwhile, the latter includes *Candida versatilis* and *C. etchellsii* and remains active until the late stage of ripening. *Z. rouxii* produces 2–4% ethanol and a small amount of glycerol from glucose, as well as 4-hydroxy-2 (or 5)-ethyl-5 (or 2)-methyl-3 (2H)- furanone (HEMF), methionol, and many other characteristic aroma constituents of soy sauce. *C. versatilis* produces characteristic aroma compounds, such as 4-ethylguaiacol (4EG), from phenolic compounds derived from wheat lignin. In the presence of a high salt concentration (18%), the pH range in which *Z. rouxii* can grow is narrowed down to 4.0–5.0, and *Z. rouxii* can vigorously grow even when the pH of the moromi mash is lowered by lactic acid fermentation. Therefore, in soy sauce brewing, three kinds of microorganisms work in stages, and multiple parallel fermentation occurs to establish the distinctive flavor of soy sauce. Halophilic lactic acid bacteria and salt-tolerant yeast are mainly present in the environment of koji, brewing containers, and brewing rooms and are mixed into the moromi mash.

However, in recent years, microbial strains with excellent fermentation performance, including lactic acid and/or alcohol fermentation, have been isolated and are now widely used in soy sauce brewing. After alcohol fermentation by yeasts, the moromi mash is allowed to mature for a certain period, and various chemical reactions take place among the diverse components of the moromi. For example, amadori compounds are formed by Maillard reactions between sugars, such as glucose, and amino acids produced by the enzymatic decomposition of raw materials. These compounds are responsible for the distinctive color, mildness, and umami taste of soy sauce. Nearly 300 aromatic compounds have been identified in soy sauce, the majority of which are formed during fermentation and aging.

After fermentation and aging, the moromi mash is wrapped in a square cloth, weighed, and continually pressed. The liquid content produced is raw soy sauce, which undergoes heat treatment for pasteurization and enzyme inactivation. During the heating process, enzymes are denatured, causing particles to coagulate and settle out. Heat also accelerates the chemical reactions involved in soy sauce brewing, thereby contributing to its unique flavor. In addition, only the clear part of the supernatant is extracted to produce the final product. Soy sauce brewing lasts approximately six months.

## 3. Koji Mold Species

Koji mold used in soy sauce brewing is a filamentous fungus belonging to the genus *Aspergillus* and is roughly classified into three species: *Aspergillus oryzae*, *Aspergillus sojae*, and *Aspergillus tamarii*. *A. oryzae* is widely used in the manufacture of sake, amazake (Japanese sweet beverage), mirin (Japanese sweet rice wine), miso, and soy sauce, whereas *A. sojae* is only used in miso and soy sauce production (mostly in soy sauce). Meanwhile, *A. tamarii* is used very infrequently. Therefore, the main koji molds used for soy sauce brewing include *A. oryzae* and *A. sojae* [4].

*A. oryzae* was first isolated from rice koji by H. Ahlburg in 1876 [5]. The morphological characteristics of the conidia were rough, and a few were echinulate. Sakaguchi et al. isolated *A. sojae* as distinguishable from *A. oryzae* in koji used for soy sauce production in Japan and found that *A. sojae* produced prominently echinulate conidia and smooth-walled conidiophores. [6].

*A. oryzae* and *A. sojae* were selected and bred as strains suitable for soy sauce brewing. Therefore, it is presumed that *A. flavus* and *A. parasiticus*, which are naturally occurring aflatoxin-producing pathogenic fungi, are domesticated. Recently, comparative genomic analysis results have suggested that *A. oryzae* is taxonomically closer to *A. aflatoxiformans* than to *A. flavus,* and that *A. aflatoxiformans* is the ancestor of *A. oryzae* [7,8]. The conidiophore of selected characteristic koji molds is usually supplied as seed koji (koji starter) to sake, miso, and soy sauce brewers; however, several large manufacturers of soy sauce in Japan produce their own koji mold in-house.

### 3.1. Differences between A. sojae and A. oryzae

The main role of koji mold in soy sauce brewing is to produce various enzymes that decompose large molecular substances, such as proteins and starches, in the raw materials into smaller substances, such as amino acids and glucose.

One main difference between *A. oryzae* and *A. sojae* is their α-amylase and endopolygalacturonase productivity. *A. oryzae* has higher α-amylase productivity, whereas *A. sojae* exhibits higher endopolygalacturonase productivity [9]. Owing to its high amylolytic activity that produces more glucose and subsequently promotes alcoholic fermentation in yeast, *A. oryzae* is widely in soy sauce, miso, and sake brewing. According to the comparative analysis of their genome sequences, *A. oryzae* RIB40 possesses three amylase genes, whereas *A. sojae* NBRC4239 only contains one amylase gene [10]. This result has also been confirmed in other industrial *A. sojae* strains. The disruption of this gene results in the loss of α-amylase activity [11]. In addition, the amylase promoter of *A. sojae* has a base substitution (CCAAT ≥ CCAAA) in the CCAAT sequence that binds the CCAAT-binding proteins, which is essential for the strong transcriptional activation of the α-amylase genes (*amyB* and *amyC*) of *A. oryzae* RIB40, and its promoter ability is also reduced [11]. In soy sauce brewing, excessively high α-amylase activity is undesirable as high sugar consumption of koji mold during koji making slows down subsequent lactic acid and yeast fermentation. Therefore, *A. sojae* is more suitable for brewing soy sauce.

There are some differences in the characteristics of soy sauce produced by *A. sojae* and *A. oryzae* in the laboratory scale [12]. For instance, owing to citric acid assimilation by *A. sojae*, the pH of soy sauce koji it produces is higher due to lower citric acid content. In addition, *A. sojae* exhibits low carbohydrate consumption during koji making, significantly high endopolygalacturonase and glutaminase activities, and low α-amylase, acid protease, and acid carboxypeptidase activities. The viscosity of moromi mash produced by *A. sojae* is also low, as well as the amount of coagulation and sedimentation after heat treatment of raw soy sauce. Lastly, the concentrations of reducing sugars, lactic acid, and glutamic acid per total nitrogen component in raw soy sauce fermented using *A. sojae* are high. These differences are almost identical in soy sauce produced on a factory scale and can be distinguished between soy sauce made with *A. oryzae* and that with *A. sojae* by panelists using sensory evaluation [13].

Furthermore, there are more volatile compounds in soy sauce produced by *A. sojae* than *A. oryzae*, particularly ethyl lactate, acetic acid, pyrazine, phenylacetaldehyde, phenol, and maltol. In contrast, 2-methyl-1-butanol, 3-methyl-1-butanol, 2-phenylethanol, 2-methylbutanoic acid, 3-methylbutanoic acid, 3-methylthio-1-propanol, 2-ethyl-4-hydroxy-5-methyl-3 (2H)-furanone, 4-ethylguaiacol, and 4-ethyl phenol are more abundant in the soy sauce from *A. oryzae*. These differences in aromatic components may affect the sensory properties of the final products [14].

The amount of methyl maltol in soy sauce koji made from *A. sojae* is larger than that from *A. oryzae*. Methyl maltol is converted from maltol in soybeans by koji mold during the koji production process and is directly transferred to raw soy sauce. Focusing on the difference in the productivities of methyl maltol by both strains, the *Aspergillus* species used was identified from the amount of methyl maltol in raw soy sauce. The majority of soy sauce products in Japan are manufactured using *A. oryzae* [15].

If the phytic acid contained in soybean remains undecomposed in raw soy sauce, it can cause severe turbidity during heating. At 30 d after brewing, phytic acid was not detected in soy sauce made with *A. sojae*, whereas some remained in that with *A. oryzae*. However, there was no difference in phytase activity between these koji strains. Thus, this difference in phytic acid level is attributed to the lower thermostability of phytase in *A. oryzae*. To reduce the phytic acid content of soy sauce, brewing at low-temperature using *A. oryzae* or using a mixture of *A. sojae* and *A. oryzae* has been proposed [16].

Lactic acid fermentation sometimes occurs at insufficient levels in soy sauce prepared with *A. oryzae*. This is attributed to the production of heptelidic acid (HA), an antibacterial compound, by *A. oryzae* [17]. *T. halophilus* requires a large amount of ATP to maintain salt tolerance; however, HA inhibits the glyceraldehyde-3-phosphate dehydrogenase (GAPDH) activity of *T. halophilus*, leading to intracellular ATP depletion and growth inhibition. However, some strains of *T. halophilus* have amino acid substitutions in GAPDH and are thus resistant to HA. Hence, adding *T. halophilus*—resistant to HA—is necessary for the soy sauce brewing process using *A. oryzae*. The gene cluster responsible for the biosynthesis of HA in *A. oryzae* RIB40 has been identified. *A. sojae* NBRC4239 lacks this gene cluster and therefore does not produce HA [18].

### 3.2. Desirable Characteristics for Soy Sauce Brewing

The desirable characteristics of koji mold for soy sauce brewing are as follows: (1) high productivity of enzymes, especially proteolytic enzymes, glutaminase, and macerating enzymes; (2) short stalk length and excellent conidiation ability; (3) rapid growth rate and high mycelial content; (4) low consumption of starch during growth; (5) genetically stability; and (6) absence of aflatoxin biosynthesis and little or no production of other mycotoxins. However, selecting a strain exhibiting all these properties is difficult as koji molds have extremely diverse characteristics. Therefore, strains are sometimes bred by mutagenesis to enhance their abilities or modify undesirable characteristics. For example, *A. sojae* KS was irradiated with X-rays to obtain a mutant strain, X-816, which was twice as potent as the parent strain in the production of proteases [19]. X-816 was further irradiated with ultraviolet light (UV) to enhance its protease production [20]. Moromi produced from *A. sojae* No. 262 strain, whose glutaminase productivity was enhanced by UV irradiation, had increased glutamate content compared to its parent strain [21]. In addition, using a mutant strain with double markers for conidial color and auxotrophy, a diploid strain exhibiting the intermediate enzyme productivity of its two parental strains was generated by protoplast cell fusion of *A. sojae* 2048 strain with high protease activity and *A. sojae* 2165 strain with high glutaminase activity [22]. These diploid strains were irradiated with UV and treated with benomyl for haploidization, thus obtaining strains with high protease and glutaminase activities [23].

Mutagenesis by ion beam irradiation has also been employed as a mutation breeding technology for koji mold [24]. For example, *A. oryzae* RIB40 was irradiated with gamma rays and two types of carbon ion beams (^12^C^5+^, ^12^C^6+^), and mutagenic effects were evaluated by monitoring the frequency of selenate-resistant mutants. A high mutant frequency was observed in a dose range that resulted in a survival rate ranging between 0.1% and 1% for all types of radiation. The highest mutation frequency was observed with a ^12^C^6+^ ion beam at a dose of 700 Gy; however, the ^12^C^5+^ ion beams were most effective in mutation frequencies at low doses. In addition to base substitutions, selenate-resistant mutants obtained by ion beam irradiation showed significant changes in chromosome structure, such as large-scale deletions and translocations. Moreover, *A. sojae* was irradiated with carbon ion beams, and the mutation frequency was evaluated by monitoring the frequency of pyrithiamine-resistant mutants. The highest mutation frequency was reported using carbon ion beams at a dose that resulted in a survival rate of less than 1%. Because ion beam irradiation causes large-scale structural changes in chromosomes, this method is expected to produce novel mutant strains.

Recently, a new mutagenesis technique using atmospheric and room-temperature plasma (ARTP) system developed in China was employed to breed koji molds. *A. oryzae* 3.042, which was generated by repeated UV irradiation, as the parental strain, was treated with ARTP for 180 s, to obtain the H8 strain with high protease activity, including that of neutral protease, alkaline protease, and aspartate aminopeptidase [25]. Soy sauce produced from H8 strain showed no difference in total free amino acid content; however, its peptide content of 1–5 kDa was significantly higher than that of soy sauce from the parent strain. These peptides may contribute to the enhancement of the umami and kokumi taste. Furthermore, derived from the conidia of *A. oryzae* 3.042 irradiated with ARTP for 150 s, the B-2 mutant exhibiting genetically stable acidic protease activity even after 10 generations of passage was obtained [26].

### 3.3. Nonaflatoxigenicity

*A. oryzae* and *A. sojae* do not produce the carcinogen aflatoxin (AF). The non-productivity of aflatoxin has been demonstrated by numerous studies as materially undetectable and not biosynthesized. *A. sojae* harbors a 70-kb aflatoxin biosynthetic gene cluster in its genome. However, there is a nonsense mutation in the transcription factor AflR, which positively regulates gene expression in the AF biosynthetic gene cluster, resulting in the deletion of 62 amino acids on the C-terminal side, including the transcriptional activation region. This results in the loss of transcriptional activation, and genes involved in aflatoxin biosynthesis are not expressed [27]. This nonsense mutation is also conserved in other *A. sojae* strains derived from soy sauce, which were misclassified as *A. oryzae* [28]. In addition, *pksA*, the polyketide synthase gene that synthesizes an important intermediate in aflatoxin biosynthesis, is defective due to the pre-termination stop codon, and even if a functional AflR is expressed, no aflatoxin is produced [29]. These nonsense mutations have also been confirmed in the draft genome sequence of *A. sojae* NBRC4239.

Recently, 14 strains of *A. sojae* have been isolated from Meju, a traditional fermented soybean brick that is produced in different regions in Korea. Nonsense mutations in *aflR* and *pksA* were identified in two of the strains, i.e., SMF127 and SMF131, which showed high proteolytic activity and were candidates for fermentation starters [30]. According to whole-genome sequencing analysis, these nonsense mutations are also conserved in *A. sojae* SMF134 [31], suggesting that they are also conserved in *A. sojae* strains derived from fermented foods other than soy sauce.

Approximately half of *A. oryzae* strains used for the production of Japanese fermented foods lack the AF biosynthetic gene cluster, and even if the cluster is almost completely retained in the genome, the expression of the *aflR* gene is extremely low [32]. Furthermore, there is a six-base difference in the promoter region of AflR (also shared by AflJ) between *A. oryzae* and aflatoxigenic *A. flavus*, and four-amino acid substitution mutation in AflJ is conserved in *A. oryzae* [33]. Even if *aflR* is expressed, AflJ, which interacts with AflR to regulate the AF biosynthetic gene cluster, remains defective, and AflR fails to function [32,33]. Overall, Koji molds used in the manufacture of Japanese fermented foods employ multiple strategies to block aflatoxin biosynthesis, thereby ensuring food safety.

## 4. Koji Mold Enzymes for Soy Sauce Brewing

The distinctive flavor of soy sauce is attributed to the interplay between taste, aroma, and color. Soy sauce contains all five basic tastes: sweet, salty, sour, bitter, and umami (savory), and the harmonious combination and subtle balance of which contributes to its application as all-purpose seasoning. Sweetness comes from sugars, such as glucose, saltiness from salt, and sourness from organic acids, such as lactic and acetic acids. However, sweetness, bitterness, and umami are also derived from amino acids and peptides, which are degradation products of raw material proteins. Sweet amino acids, such as glycine, contribute to sweetness; branched-chain amino acids, such as leucine and peptides, contribute to bitterness; and the 20 amino acids themselves, including acidic amino acids such as glutamic acid and aspartic acid, greatly contribute to the umami taste of soy sauce. Because amino acids and peptides are crucial to the taste of soy sauce, the degradation and solubilization of insoluble raw material proteins have been extensively studied to improve the flavor and yield of soy sauce.

### 4.1. Proteolytic Enzymes

The insoluble proteins of soybeans and wheat are first broken down into large and small peptides and degraded and solubilized into various amino acids by proteolytic enzymes of koji mold. In the soy sauce industry, solubilized nitrogen compounds, such as amino acids, are termed total nitrogen (TN) and are used for quality and grading. Proteolytic enzymes are classified into endopeptidases, which hydrolyze peptide bonds inside proteins and peptides, and exopeptidases, which hydrolyze peptide bonds at the ends of proteins and peptides. The former hydrolyzes proteins into large peptides, whereas the latter hydrolyzes proteins into smaller peptides, such as dipeptides, tripeptides, and amino acids.

Endopeptidases are further classified into six types based on the structure of the active site: serine, cysteine, aspartic, metal, threonine, and unclassified. Meanwhile, exopeptidases are classified according to their mode of action, i.e., whether they act at the N- or C-terminus. *A. oryzae* RIB40 and *A. sojae* SMF134 have 65 and 83 endopeptidase genes and 69 and 67 exopeptidase genes in their genome, respectively [31].

As the degradation of soybean proteins, which are the main protein source in soy sauce, influences the quality (taste) of soy sauce, the proteolytic enzymes of koji mold have been extensively studied, and many of which have been isolated and purified. From the degradation studies of soybean α-protein using purified enzymes, the contribution rate of the enzyme involved in the solubilization of soybean protein in soy sauce brewing was evaluated. 

In *A. oryzae*, three endopeptidases (alkaline protease and neutral proteases I and II) and two peptidases (leucine aminopeptidases I and II) contribute to the solubilization of raw material proteins in soy sauce brewing [34]. In *A. sojae*, the same three endopeptidases in *A. oryzae* and three peptidases (leucine aminopeptidases I and III and acid carboxypeptidase IV) have been shown to contribute to the degradation of raw material proteins in soy sauce brewing [34]. Several leucine aminopeptidases and acid carboxypeptidases have also been reported to contribute to the release of glutamic acid in soy sauce [34,35].

### 4.2. Alkaline Proteinase

In soy sauce koji, alkaline proteinase activity is extremely high, and this enzyme is deeply involved in the solubility of soy sauce moromi. The extracellular alkaline proteinase (Alp) of koji mold is a serine proteinase, also known as Oryzin. The Alp from *A. sojae* has an optimum pH of 11 [36], whereas that from *A. oryzae* ranges from pH 7 to 10.5 [37]. However, under the same conditions (substrate, buffer solution, etc.), there was no difference in their enzymatic properties (such as optimal pH and pH stability). Moreover, a wide range of optimal pH (7–10) has been observed for both strains [38]. Protein hydrolysis ceases when the degree of peptide bond hydrolysis reaches 9–15%. Amino acid liberation is rarely observed during this process. Alkaline protease has a relatively high specificity for hydrophobic (leucine), aromatic, and basic amino acid residues on the carboxyl side of the peptide linkage at the cleavage point [39].

The gene encoding alkaline protease was cloned as cDNA from the mRNA of *A. oryzae* ATCC20386 and expressed in *S. cerevisiae*. The extracellular recombinant protein was purified, and various properties of the enzyme, including molecular weight, specific activity, optimal pH, N-terminal amino acid sequence, and no glycosylation, were found to be similar to those of *A. oryzae* [40].

AlpB (AO090020000517) was identified in the genome sequence of *A. oryzae* RIB40 as a homologous gene of alkaline protease (AlpA; AO090003001036). The alkaline protease activity of soy sauce koji prepared using the *alpA* disruptant was significantly reduced. On the other hand, soy sauce koji made by the *alpB* disruptant has significantly higher protease activity than that by the parent strain, although the ability of conidiation was significantly reduced [41].

### 4.3. Neutral Proteinase

The proteolytic activity of soy sauce koji extract on milk casein was more than 90% of the total activity occupied by alkaline protease; however, on soy protein, the contribution of alkaline protease was 50%. In addition, the proteolytic activity on milk casein in the presence of ethylenediamine tetra acetic acid (EDTA) was not significantly inhibited, whereas that on soybean protein was inhibited by approximately 50% [42]. On the basis of these findings, a neutral protease was isolated.

Neutral proteases (Np) are zinc-containing metalloproteases with an optimal pH in the neutral range (pH 6.0–7.0). There are two types of Np: neutral protease I (Np I) and II (NpII) also known as deuterolysin. Np I effectively digested various proteins, particularly milk casein and soybean proteins, whereas Np II hardly digested these substrates and showed specificity for basic proteins, such as histones and protamines. Moreover, NpII had a narrower substrate specificity than Np I [43]. On the other hand, Np II is more thermostable than Np I and retains its activity even after treatment at 99 °C for 10 min [44]. The Np II of *A. sojae* became unstable after treatment at 75 °C due to autoproteolysis, but this was not observed in that of *A. oryzae* [45]. However, Np II from the *A. oryzae* ATCC20386 strain showed the same properties as that from *A. sojae* [46]. Therefore, the property of Np II at 75 °C may differ among strains. Both enzymes digest proteins but did not produce low-molecular-weight peptides or amino acids.

The Np I gene of *A. oryzae* 3.042 strain has been expressed in *Pichia pastoris* and possesses an optimal pH range of 7.0–8.0 and a stable pH in the neutral range (5.0–9.0). Thus, its activity was rapidly lost by heat treatment above 55 °C, which hydrolyzes proteins from soybeans and peanuts better than papain [47].

Two Np II genes, *deuA* (NpIIa) and *deuB* (NpIIb), from *A. oryzae* RIB40 were expressed in *A. oryzae* [48]. The *deuA* gene is identical to the cloned gene as an NP II from *A. oryzae* ATCC20386 [49]. DeuA was stable above 80 °C, whereas DeuB lost its activity at 80 °C but had a small amount of activity above 80 °C. Both enzymes showed high specificity for basic proteins such as clupeine and salmin and low reactivity for other proteins. Interestingly, the *deuA* gene was not expressed in the liquid culture, even though proteinous substrates, such as skim milk, were present, suggesting that the transcriptional expression of *deuA* is solid-state culture-specific.

### 4.4. Acid Proteinase (Aspartic Protease)

Acid proteinase (AP) was isolated from *A. oryzae* 460 with an optimum pH of 3.7 and a molecular weight of 39,000 Da. Upon the hydrolysis of α-soybean protein by the purified acid proteinase, peptides consisting of 7–9 amino acid residues were detected; however, no amino acids were released. On the other hand, upon its hydrolysis by acid protease with carboxypeptidase IV at 30 °C and pH 5, glutamic acid was detected, and the amount of free amino acids increased [50]. Therefore, acid proteases are involved in the degradation of raw material proteins in soy sauce brewing.

The *pepA* gene encoding as an acid proteinase has been cloned from *A. oryzae* RIB40 [51]. The same gene has been cloned from another *A. oryzae* strain as the *pepO* gene [52]. The *pepA* gene was specifically expressed in solid-state cultures but not in submerged cultures and exhibited a temperature-dependent expression, which is suppressed at high temperatures [53].

The total number of aspartic protease genes in *A. sojae* was higher than that in *A. oryzae* [31]. Moreover, the aspartic protease gene unique to *A. sojae* is present in the genome of *A. sojae* NBRC4239, confirming its expression in wheat bran medium [10]. This gene has also been successfully expressed in *S. cerevisiae* and *A. sojae* and has been shown to function as an acid proteinase [54].

### 4.5. Leucine Aminopeptidase

Leucine aminopeptidase (LAP) is a metal exopeptidase that selectively removes an amino acid residue, primarily leucine, from the N-terminus of peptides and proteins. In *A. oryzae* used in soy sauce production, four extracellular LAPs (I, II, III, and IV) have been purified and characterized. These LAPs have an optimum pH range of 7.0–8.5 and are less likely to hydrolyze peptides with glycine or acidic amino acids at the N-terminus [55,56].

Genes for LAP I have been identified and characterized in *A. oryzae* and *A. sojae* [57,58]. The LAP I of *A. oryzae* RIB40 can release hydrophobic amino acids, such as Leu and Phe, and basic amino acids from its N-terminus. However, its reactivity to glutamic acid, an acidic amino acid, is extremely low; thus, it cannot hydrolyze aspartic acid. Furthermore, it is almost inactive against dipeptides and tripeptides, requiring peptides longer than tripeptides. The LAP I of *A. oryzae* and *A. sojae* showed a high amino acid identity (97%), but their substrate specificities, such as reactivity to phenylalanine, were slightly different.

LAP II was purified from *A. oryzae* ATCC20386 and was cloned its gene [59]. The optimal pH of LAP II can be as high as 9.5–10. Unlike LAP I, its substrate specificity is very low. Although the reactivity of various para-nitroanilides of amino acids differs from that of natural peptides, LAP II is non-specific, and all amino acids can be liberated from the peptide, including Pro, Gly, and acidic amino acids. However, it cannot hydrolyze peptides with Xaa-Pro at the N-terminus.

In addition to LAP I and LAP II orthologs, LAP3, which is evolutionarily close to LAP I, has been identified in the genome of *A. sojae* SMF134 [31]. The transcriptional expression of these three genes was confirmed in the soybean solid medium. The gene encoding LAP3 was isolated from *A. sojae* ATCC42251 and expressed in *S. cerevisiae.* Its aminopeptidase activity against Leu₋p-nitroanilide has been confirmed [60].

### 4.6. Dipeptidyl Peptidase

Dipeptidyl peptidase (Dpp) releases dipeptides from the N-terminus of the protein or peptide. Particularly, DppIV excises Xaa-Pro, which is located at the N-terminal. The DppIV of koji mold was purified and characterized for the first time in *A. oryzae* RIB915 derived from soy sauce koji [61]. DppIV has an optimum pH of 7.0 to 7.5 and is a serine protease that specifically cleaves the N-terminal dipeptide, having a preference for proline at the penultimate position. Three extracellular dipeptidyl peptidases were identified in the genome of *A. oryzae* RIB40 [62], and DppB was identified as DppIV. It is difficult to release dipeptides from substrates that do not contain a Pro second position from the N-terminus. In addition, the N-terminus of the peptide is more reactive to Ala and Arg than to Gly, indicating that the N-terminal amino acid of the peptide preferably has a side chain. However, the Pro-Pro bond of the peptide whose N-terminal is Xaa-Pro-Pro cannot be hydrolyzed [62].

Soybean protein digested with crude extracts derived from wheat bran koji, prepared by separating the fraction with DPPIV activity using a gel filtration column, does not undergo low molecular weight digestion of dipeptides and amino acids, and peptides with a molecular weight of about 1000 are accumulated [63]. In addition, the amount of amino acids and Xaa-Pro production was reduced to 1/3 or less compared to the same crude extract without separating treatment containing DPPIV activity. It has been confirmed that the Xaa-Pro peptide is not hydrolyzed by LAPII, but by adding DPPIV, the constituent amino acids of the peptide are released along with Xaa-Pro [59]. The optimum pH for DPPIV of *A. oryzae* is pH 7.0 to 8.0, as is the case with LAPs. In the early stage of soy sauce fermentation, LAP II is proposed to cleave the N-terminal ends of polypeptides to sequentially produce various amino acids. Then, when they are in the form of Xaa-Pro peptides, which cannot be hydrolyzed by LAP II, DPP-IV specifically acts to release Xaa-Pro, thereby further promoting amino acid liberation by LAP II. This proposed mechanism is supported by the observed increase in Gly-Pro levels, along with other amino acids, in the early stage of soy sauce fermentation, and there are many Xaa-Pro dipeptides remaining in soy sauce [64,65,66].

### 4.7. Carboxypeptidase

Carboxypeptidase is an exopeptidase that sequentially releases amino acids from the C-terminus of peptides and proteins. It has been purified and characterized to I–VI in *A. oryzae* [67,68,69]. Its optimal pH was in the acidic range (pH 3.0–4.0). As tripeptides and dipeptides are difficult to degrade (the latter being more difficult), the ability of carboxypeptidase to liberate amino acids from peptides is somewhat slightly lower than that of leucine aminopeptidase. It is also difficult to degrade peptides that contain glycine or proline at the C-terminus.

In the genome sequence of *A. oryzae* RIB40, nine serine genes are predicted as extracellular serine-type carboxypeptidases, and gene function analysis has been performed for five of these genes (*cpI*, *ocpA*, *ocpB*, *ocpC*, and *ocpO*) [70,71,72]. All of them were stable in the acidic to neutral range, with an optimal pH of approximately 4. They have been identified as serine-type carboxylases as their activity was inhibited by PMSF and they could continuously release amino acids from the C-terminus using various small N-acylpeptides, angiotensin I, and bradykinin as substrates. Substrate specificity using various N-acylpeptides showed that the best substrate for CpI and OcpA was Z-Phe-Leu; OcpB, Z-Phe-Tyr-Leu; OcpO, Z-Tyr-Leu or Z-Phe-Leu; and OcpC, Z-Leu-Tyr. Thus, the substrate specificity of these enzymes slightly varies. All of them hardly hydrolyze Z-Gly-Pro or Z-Val-Gly, but this may be due to the length of the substrate. Among these, OcpC has an outstandingly low specific activity. These genes also differ in nutrient conditions for the induction and repression of transcriptional expression. However, the genes corresponding to carboxypeptidases I–IV, whose contribution for soy sauce brewing has been evaluated, remain unidentified.

Some serine-type carboxypeptidases are found only in *Aspergillus* section *Flavi*. A unique serine-type carboxypeptidase to *A. sojae*, which has not been reported in other *Aspergillus species,* has been found in the genome of *A. sojae* (and also *A. parasiticus*) [10,31]. Transcriptional expression of this gene has been confirmed in a wheat bran solid medium. The transformants, which this gene and the CpI ortholog gene of *A. sojae* are overexpressed in *A. sojae* under an alkaline protease promoter, are cultured in a wheat bran solid-state medium. Then, the acidic carboxypeptidase activities of the resulting crude extracts were increased 1.5-fold and 2.7-fold, respectively, compared to the control strain. In addition, the degradation of soy sauce koji using this gene-overexpressing transformant, wheat gluten, and brine resulted in an increase in the rate of degradation of amino acids compared to the control. It also showed the same effect as that using the CpI gene-overexpressing transformant [73]. This *A. sojae*-specific gene may also have an effect on soy sauce brewing.

### 4.8. Aspartyl Aminopeptidase

Aspartyl aminopeptidase (DapA) is a metalloprotease that hydrolyzes acidic amino acids located at the N-terminus of peptides [74,75]. DapA is stable between pH 5.0 to pH 8.0 with an optimal pH of 7.0–8.0. It has the highest specificity for aspartic acid, followed by glutamic acid, and showed no reactivity to other amino acids. DapA is expressed in the cytoplasm in its overexpressing transformant of *A. oryzae* in liquid culture [75], but remarkably high enzymatic activity was detected in rice koji extracts made with the transformant. It has also been identified in soy sauce koji extracts [74], indicating that DapA is easily eluted from koji mold in solid-state culture. Furthermore, DapA has moderate salt tolerance; therefore, it may be effective in soy sauce brewing. In addition, the amount of glutamic acid increases when DapA acts on defatted soybean partial degradation products [74].

### 4.9. M24B Family Peptidase (Prolidase and Xaa- Pro Aminopeptidase)

A large number of dipeptides, such as Xaa-Pro, remain in soy sauce [65]. If these dipeptides can be digested, the umami taste of soy sauce can be further improved. Therefore, two genes predicted as prolidase were found in the EST sequences of *A. oryzae* RIB40, and were expressed in *S. crevisae* in which their prolidase activities were confirmed [76,77].

The *xpmA* gene, predicted as a Xaa-Pro aminopeptidase, was isolated from *A. oryzae* RIB40, and its properties have been clarified [78]. XpmA has a wide stability range from pH 3.5 to 12.0 with an optimum pH of 8.5 to 9.0. It also hydrolyzes not only Xaa-Pro dipeptides such as Ala-Pro and Ser-Pro, which remain in soy sauce, but also oligopeptides of various lengths containing Xaa-pro at their N-terminus. However, XpmA did not act on Gly-Pro. Furthermore, XpmA shows halophilicity and salt tolerance and is expected to work well in soy sauce brewing. It has also been identified as a protein in the soy sauce koji extract made with *A. oryzae* AS 3.951, an industrial strain of Chinese soy sauce production [79].

### 4.10. Glutaminase

In 1931, monosodium l-glutamate (hereafter referred to as l-glutamate), the umami component of Konbu kelp, was found in soy sauce, and it has been shown that l-glutamate is the main component of umami in soy sauce [80].

Glutaminase (glutamine amidohydrolase, EC 3.5.1.2) catalyzes the hydrolytic deamidation of l-glutamine, resulting in the production of l-glutamate and ammonium. In soy sauce brewing, l-glutamate is produced via two pathways: (1) direct release from raw material proteins by proteases and/or peptidases and (2) the hydrolysis of free l-glutamine released from raw material proteins by glutaminases. The ratio of l-glutamine to l-glutamate contained in soy protein is approximately equal. l-glutamine is cyclized through a relatively rapid non-enzymatic reaction (chemical reaction) to pyroglutamate, which has no “umami” taste. In contrast, l-glutamate is more stable than l-glutamine in soy sauce, and the rate of nonenzymatic conversion to pyroglutamate is extremely slow. Therefore, the latter pathway is more important for enhancing the umami taste during soy sauce brewing in which glutaminase plays a key role.

The concentration of l-glutamate in soy sauce depends on the glutaminase activity of *Aspergillus* strains used for fermentation [21] and positively corresponds to the cell wall-bound glutaminase activity of *A. oryzae* [81]. Hence, l-glutamate is thought to be converted from l-glutamine by glutaminase produced by *Aspergillus* strains during soy sauce fermentation.

The intra- and extracellular glutaminase from *A. oryzae* are purified to a single band and characterized [82]. They have an optimum pH of 9.0, and hardly hydrolyze d-glutamine and d-and l-asparagine. It also hydrolyzes γ-glutamyl derivatives such as theanine, glutathione, and γ-glutamyl-p-nitroanillide, indicating the presence of γ-glutamyl transpeptidase activity. The *gtaA* gene was isolated for the first time with the glutaminase of *A. oryzae* [83]. GtaA has an optimal pH of 9.0 and γ-glutamyltranspeptidase activity, but its molecular weight is different from that of the previously known enzyme. Moreover, its substrate specificity, such as reactivity to d-glutamine, is also different. In addition, another glutaminase gene different from *gtaA* was isolated [84]. It was revealed that koji molds have multiple glutaminase genes in their genome.

A BLAST search for genes similar to the known glutaminase genes revealed the existence of 10 glutaminase genes in the genome of *A. sojae* [85], which are classified into four types based on amino acid homology: type I (GahA, GahB, and GahD), which is classified as a glutaminase-asparaginase, type II (GgtA, GgtC, and GgtD), which is classified as a γ-glutamyl transpeptidase, and type III (GtaA, GtaB, and GtaC), which is the glutaminase gene first found in koji mold, and type IV (Gls), which is homologous to bacterial glutaminase. To determine which of these 10 glutaminases contribute to the production of l-glutamate in soy sauce, single disruptants of the 10 glutaminase genes in *A. sojae* were constructed and determined the glutaminase activity of each disruptant in soy sauce koji. As a result, the glutaminase activity of the *gahB* gene disruptant decreased by approximately 90% compared to that of the control strain. This effect of *gahB* gene disruption was also confirmed in *A. oryzae*, indicating that GahB acted as the main glutaminase in soy sauce koji. However, the glutamate concentration in soy sauce prepared using the *gahB* disruptant was similar to that in soy sauce prepared using the control strain. Therefore, strains with multiple disruptions of several genes by type were constructed, and the contribution of each type of glutaminase to soy sauce brewing was evaluated by measuring and comparing the amount of glutamate per total nitrogen component in soy sauce prepared using these transformants [86]. Type I made a large contribution to glutamate production. The disruption of type I reduced glutamate by 20–30% relative to the control. In addition, the disruption of type II decreased glutamate by approximately 50%, and the disruption of type IV further decreased the amount of glutamate. Eventually, four different types of glutaminases (GahA, GahB, GgtA, and Gls) were found to be involved in glutamate production in soy sauce brewing. Gls are located intracellularly; therefore, their contribution is thought to be low. However, Gls are the only salt-tolerant among them. These may be leaked from cells that are burst by salt into the moromi and might be able to act on the released l-glutamine at a later stage of soy sauce fermentation because of its salt tolerance [87]. GgtA also exhibits γ-glutamyl transpeptidase activity. Many γ-glutamyl compounds have been detected in soy sauce [88,89]; therefore, it was proposed that γ-glutamyl compounds are hydrolyzed to produce glutamate by γ-glutamyl transpeptidase [88]. However, the amount of glutamate was at maximum even in soy sauce prepared using the triple gene disruptant, in which all type II glutaminase genes were disrupted. Therefore, glutamate is probably not derived from γ-glutamyl compounds. GahA and GahB of Type I, which contribute the most to glutamate production in soy sauce brewing, both have peptidoglutaminase activity that hydrolyzes not only free l-glutamine but also l-glutamine located at the C-terminus of the peptide to glutamic acid [85,90].

## 5. Mechanism of Glutamate (Umami) Production in Soy Sauce Brewing

A mechanism for glutamate production in soy sauce brewing has been proposed (Figure 2). Proteins derived from soybeans and wheat are degraded into peptides by proteases, such as alkaline proteases, neutral proteases, and aspartic proteases, derived from koji mold. Subsequently, these peptides produced by proteolysis are hydrolyzed into amino acids and small peptides by koji mold-derived peptidases, including leucine aminopeptidase, dipeptidylaminopeptidase, and serine-type carboxypeptidase. DPP IV acts on the Xaa-Pro peptide produced by leucine aminopeptidase to remove Xaa-Pro and enhance the activity of leucine aminopeptidase. Peptides with acidic amino acid residues at the N-terminus may be hydrolyzed by DapA.

In addition, glutamate production can occur via three different pathways. In the first pathway, l-glutamate and l-glutamine can be directly released from the raw material proteins (pathway A) as they are among the constituent amino acids of the protein components. In the second pathway, the released l-glutamine can be converted to glutamate by glutaminases (pathway B). In this pathway, four glutaminases (GahA, GahB, GgtA, and Gls) of koji mold are involved. The early stage of moromi fermentation is characterized by a neutral pH (pH 6–7). At this stage, the four glutaminases efficiently work with alkaline protease, neutral protease, and leucine aminopeptidase, all of which have a high pH optimum. In the third pathway, the C-terminal glutamine residue of the peptide is deamidated to glutamate by the peptidoglutaminase reaction of GahA and GahB proteins, which are then released by peptidase (pathway C). Glutaminases other than Gls, which contribute to glutamate production in soy sauce brewing, have a high optimum pH and poor salt tolerance. By acting on the C-terminal glutamine of the peptide and converting it to glutamate before glutaminase (peptidoglutaminase) is inactivated, leucine aminopeptidase slowly breaks down the peptide and releases glutamate even after glutaminase is inactivated. The pH of soy sauce gradually decreases due to the amino acids released from the raw protein and the organic acids, such as lactic acid and acetic acid, produced by *T. halophilus* during fermentation. In addition, serine-type carboxypeptidases, which have an acidic optimum pH, become activated to release glutamate from peptides whose C-termini are converted to glutamate. Therefore, the peptidoglutaminase reaction, which converts the C-terminus of peptides to glutamate in the peptide state, is a key factor for enhancing glutamate content in soy sauce brewing.

## 6. Conclusions

*A. oryzae* and *A. sojae* are the two main koji mold species employed for soy sauce brewing. because these species exhibit different properties, they yield soy sauce products with different qualities. In the soy sauce brewing process, various proteolytic enzymes with different substrate specificities solubilize and break down insoluble soy proteins into amino acids via peptides. Koji mold contains many more proteolytic enzymatic genes than those we discussed here, and these genes may also be involved in soy sauce brewing. To date, the genome sequences of 97 strains of *A. oryzae* and 5 strains of *A. sojae* have been reported. Genes unique to each strain have also been identified.

This review focuses on the umami-producing enzymes involved in soy sauce brewing. Many other enzymes related to plant cell wall degradation, such as pectinolytic enzymes, xylanolytic enzymes, cellulolytic enzymes, and ferulic acid esterases, are involved in the degradation of raw materials and the production of flavors in soy sauce brewing, and several genes of which have been identified. Although many enzymatic genes have been identified, it remains unclear which enzymes play an important role in soy sauce brewing. In soy sauce fermentation, the enzymes of koji mold participate in the coagulation and sedimentation of raw soy sauce by heating, and their overproduction leads to increased amounts of coagula in the final product, resulting in a lower yield of soy sauce. Ideally, the koji mold should produce a quotiety of enzymes essential for soy sauce production. With the technological advancements in genetic engineering, it is expected that more key enzymatic genes in soy sauce brewing will be identified and that new koji mold strains for the production of these crucial enzymes will be developed, thus contributing to the improvement of soy sauce productivity.

## Figures and Tables

**Figure 1 jof-07-00658-f001:**
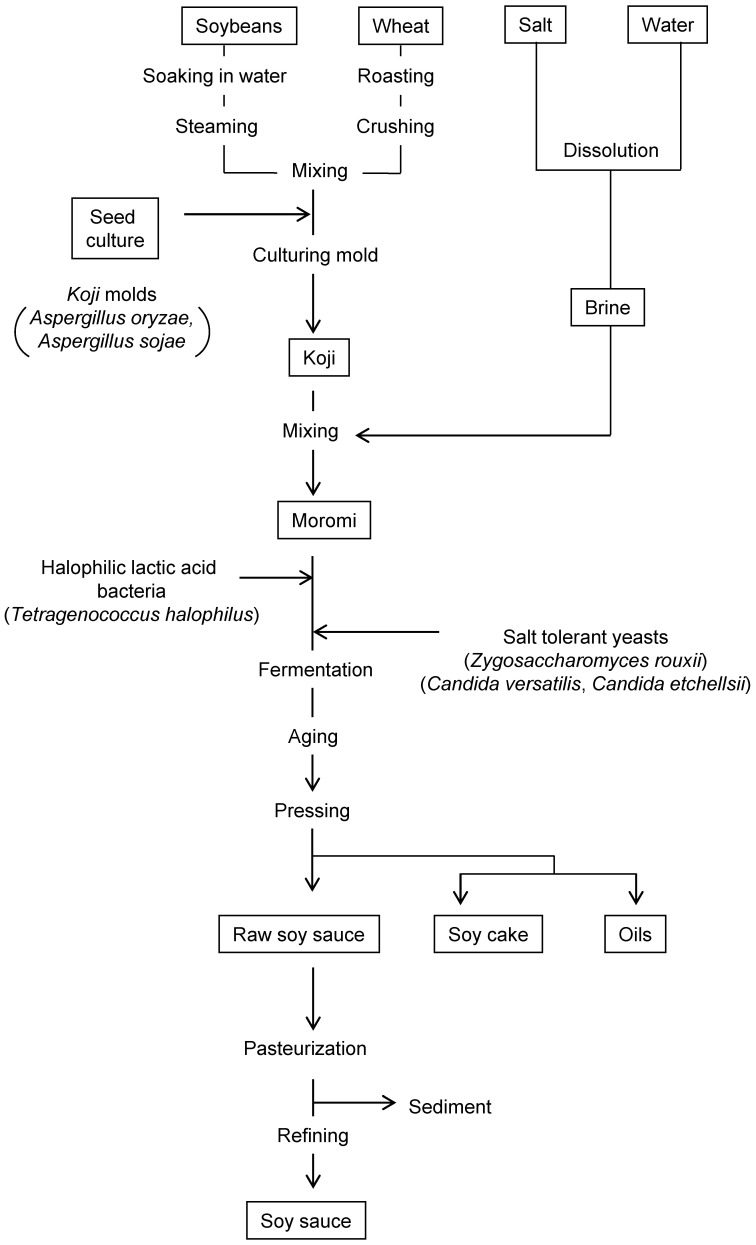
Traditional manufacturing process of soy sauce based on “honjozo,” a Japanese brewing method.

**Figure 2 jof-07-00658-f002:**
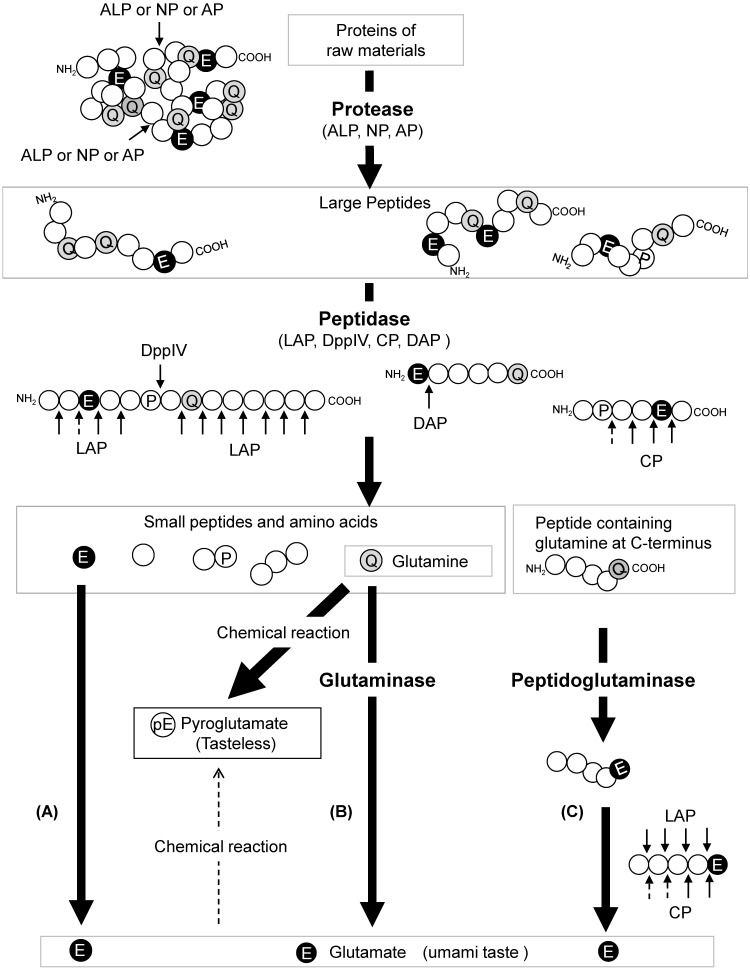
Mechanism of glutamate production in soy sauce fermentation. L-glutamate is produced via three pathways: (**A**) direct release from raw material proteins by proteases and/or peptidases, (**B**) glutaminase-mediated hydrolysis of free l-glutamine released from material proteins, and (**C**) release from peptidyl C-terminal l-glutamate residues that are deamidated by peptidoglutaminase. Pyroglutamate is non-enzymatically converted from l-glutamine and l-glutamate, but l-glutamine is converted to pyroglutamate much more efficiently than l-glutamate. The dotted arrows indicate the low efficiency of protease hydrolysis and chemical conversion. ALP, alkaline proteases; NP, neutral proteases; AP, aspartic proteases; LAP, leucine aminopeptidase; DppIV, dipeptidyl aminopeptidase; DAP, aspartyl aminopeptidase; CP, carboxyl peptidase. The circles in the figure represent amino acids: E for glutamic acid, Q for glutamine, pE for pyroglutamic acid and P for proline.

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
