# Peer review of "Koji Molds for Japanese Soy Sauce Brewing: Characteristics and Key Enzymes"

_jof, 2021, doi:10.3390/jof7080658_

Round 1

Reviewer 1 Report

This review paper is well written and would be informative to readers who are interested in soy sauce fermentation. However, I suggest authors to consider some revisions described below.

1. Title

Soy sauce is manufactured in East Asia by different ways and various molds are involved in its fermentation. Because this manuscript is limited to Japanese soy sauce and other molds in addition to molds mentioned in this paper are involved in other soy sauce fermentation, the title of “Koji molds for soy sauce brewing…” should be changed to “Koji molds for Japanese soy sauce….”

2. Line 127

According to the reference 31, several A. sojae have been isolated form meju collected from different regions. Because traditional meju is manufactured by fermentation of naturally inoculated microorganisms, it is suggested that A. sojae is present in a natural niche.

3. Line 277 and 4. 10 Glutaminase

According to the reference 89, soy sauce exhibits kokumi taste imparted by gamma-glutamyl peptides (GGP). Therefore, gamma-glutamyl transpeptidase which is involved in GGP production should be described in more detail in the separated section because it is an important enzyme for soy sauce flavor.

Author Response

Dear reviewers

We thank reviewers for careful reading our manuscript and for giving useful comments. In response to the Reviewers' comments, we have revised the manuscript. We look forward to a publication of our manuscript in Journal of Fungi.

Our responses to the reviewer's comments are in red as follows:

Response for the Reviewer #1

This review paper is well written and would be informative to readers who are interested in soy sauce fermentation. However, I suggest authors to consider some revisions described below.

  1. Title

Soy sauce is manufactured in East Asia by different ways and various molds are involved in its fermentation. Because this manuscript is limited to Japanese soy sauce and other molds in addition to molds mentioned in this paper are involved in other soy sauce fermentation, the title of “Koji molds for soy sauce brewing…” should be changed to “Koji molds for Japanese soy sauce….”

⇒Thank you for your comments. We change the title as your suggestion.

  1. Line 127

According to the reference 31, several A. sojae have been isolated form meju collected from different regions. Because traditional meju is manufactured by fermentation of naturally inoculated microorganisms, it is suggested that A. sojae is present in a natural niche.

⇒Thank you for your comment. It is believed that it is not isolated from natural soil because it is only isolated from the brewing environment, a special environment different from the nature. However, in accordance with your comment, Line 125 will be removed.

  1. Line 277 and 4. 10 Glutaminase

According to the reference 89, soy sauce exhibits kokumi taste imparted by gamma-glutamyl peptides (GGP). Therefore, gamma-glutamyl transpeptidase which is involved in GGP production should be described in more detail in the separated section because it is an important enzyme for soy sauce flavor.

⇒Thank you for pointing this out. The kokumi substances (e.g. γ-EVG) have been found in soy sauce, and some glutaminases of Aspergillus sp. have γ-glutamyl transpeptidase activity. However, which gene is involved is not clear, so I have not listed it. In addition, It has been reported that yeast also produces γ-EVG, and it is unclear whether the kokumi substance identified in soy sauce is due to the enzyme of koji mold. Do I need to write a separate section with more details? Especially this time, I am not mentioning the kokumi taste. We are just saying that in soy sauce brewing, there is no pathway for glutamate production via gamma-glutamyl compounds.

Reviewer 2 Report

I have read carefully the manuscript entitled "Koji molds for soy sauce brewing: characteristics and key enzymes." and found it to be an interesting study on the production and activity of enzymes isolated from koji molds. The authors presented a complete presentation of the stages of soybean production, analyzed various groups of microorganisms used at different stages of production, and described the activity of mainly proteases and amylases isolated from koji molds. The presented manuscript meets all the requirements for a review and has a high cognitive value. However, there are some improvements (only minor) that should be corrected before publication.

# 1: Authors should better emphasize the novelty of the presented issue in relation to the works already published on a similar subject.

# 2: In Figure 1, the full names of the microorganisms should be added.

# 3: Line 150. Please add an explanation of the abbreviation (HAP). Please check the entire manuscript.

# 4: 3.3. Nonaflatoxigenicity section: Do A. oryzae and A. sojae strains produce any type of aflatoxin B1, B2, G1 and G2?

# 5: 3.3. Nonaflatoxigenicity section: Has the influence of koji production conditions on the amount of aflatoxins produced been analyzed?

# 6: The authors mainly focused on amylases and proteases produced by koji fungi (the activity of phytases has also been described), but it may be worth adding information about other enzymes activities isolated from this group of microorganisms.

Author Response

Dear reviewers

We thank reviewers for careful reading our manuscript and for giving useful comments. In response to the Reviewers' comments, we have revised the manuscript. We look forward to a publication of our manuscript in Journal of Fungi.

Our responses to the reviewer's comments are in red as follows:

Response for the Reviewer #2

I have read carefully the manuscript entitled "Koji molds for soy sauce brewing: characteristics and key enzymes." and found it to be an interesting study on the production and activity of enzymes isolated from koji molds. The authors presented a complete presentation of the stages of soybean production, analyzed various groups of microorganisms used at different stages of production, and described the activity of mainly proteases and amylases isolated from koji molds. The presented manuscript meets all the requirements for a review and has a high cognitive value. However, there are some improvements (only minor) that should be corrected before publication.

# 1: Authors should better emphasize the novelty of the presented issue in relation to the works already published on a similar subject.

⇒Thank you for your comment. In this review, we introduced the manufacturing process of Japanese soy sauce and summarized the differences between the two types of koji molds used in soy sauce brewing, including the enzymes involved in soy sauce quality. Previous reviews have only introduced enzymes, but few have connected the relationship between enzymes and their encoding genes. Therefore, we focus on the enzymes involved in umami production in soy sauce brewing and the genes that encode them. I have added a sentence to that effect in line 38 to 42.

# 2: In Figure 1, the full names of the microorganisms should be added.

⇒I rewrote the full name of the microorganisms as your comments.

# 3: Line 150. Please add an explanation of the abbreviation (HAP). Please check the entire manuscript.

⇒I can't find the list of abbreviation in other papers in the Journal of Fungi. In another paper, I found the following: the CCAAT-binding complex in Aspergillus species, known as the Hap complex. Therefore, we changed “HAP complex” to “the CCAAT-binding proteins” in line 149.

# 4: 3.3. Nonaflatoxigenicity section: Do A. oryzae and A. sojae strains produce any type of aflatoxin B1, B2, G1 and G2?

⇒Neither A. oryzae nor A. sojae produce aflatoxins.

# 5: 3.3. Nonaflatoxigenicity section: Has the influence of koji production conditions on the amount of aflatoxins produced been analyzed?

⇒It has been analyzed under the growth conditions of soy sauce brewing. Koji molds don’t produce aflatoxins under any culture condition.

# 6: The authors mainly focused on amylases and proteases produced by koji fungi (the activity of phytases has also been described), but it may be worth adding information about other enzymes activities isolated from this group of microorganisms.

⇒Thank you for your comment. As you mention, other enzymes are also involved in soy sauce brewing. However, writing everything would be quite long, so this time I focused on umami taste involved in soy sauce. The reason I mentioned amylase and phytase is only because the difference between A. oryzae and A. sojae is clear.